

# Analysis of genetic diversity and population structure of *Magnaporthe grisea*, the causal agent of foxtail millet blast using microsatellites

Manimozhi Dhivya[1], Govindasamy Senthilraja[2], Nagendran Tharmalingam[3,4], Sankarasubramanian Harish[2], Kalaiselvan Saravanakumari[2], Theerthagiri Anand[2] and Sundararajan Thiruvudainambi[1]

[1] Department of Plant Pathology, Agricultural College and Research Institute, Tamil Nadu Agricultural University, Madurai, Tamil Nadu, India
[2] Department of Plant Pathology, Tamil Nadu Agricultural University, Coimbatore, Tamil Nadu, India
[3] Infectious Disease/Medicine, The Miriam Hospital/Rhode Island Hospital, Brown University, Providence, RI, USA
[4] Department of Medicine, Houston Methodist Research Institute, Houston, TX, USA

## ABSTRACT

Foxtail millet blast caused by *Magnaporthe grisea* is becoming a severe problem in foxtail millet growing regions of India. The genetic diversity and population structure of foxtail millet infecting *M. grisea* is crucial for developing effective management strategies, such as breeding blast-resistant cultivars. We analyzed thirty-two *M. grisea* isolates from ten foxtail millet-growing districts in Tamil Nadu, India for genetic diversity using twenty-nine microsatellite or simple sequence repeat (SSR) markers. A total of 103 alleles were identified with a mean of 3.55 alleles/locus. Gene diversity ranged from 0.170 to 0.717, while major allelic frequencies ranged from 0.344 to 0.906. The polymorphism information content (PIC) ranged from 0.155 to 0.680, with a mean value of 0.465. Population structure analysis of the genomic data sets revealed two major populations (SP1 and SP2) with different levels of ancestral admixture among the 32 blast isolates. Phylogenetic analysis classified the isolates into three major clusters. Analysis of molecular variance (AMOVA) showed high genetic variation among individuals and less among populations. Principal Coordinate Analysis (PCoA) revealed 27.16% genetic variation among populations. The present study provides the first report on the genetic diversity and population structure of the foxtail millet-infecting *M. grisea* population in Tamil Nadu, which could be useful for the development of blast-resistant foxtail millet cultivars.

# INTRODUCTION

Foxtail millet (*Setaria italica* (L.) P. Beauv.) is one of the most important climate-resilient nutri-cereal crops that comprise a vast array of nutritional properties such as protein, starch, fiber, minerals, and antioxidants. It is widely cultivated mostly in tropical and sub-tropical countries (*Sharma & Niranjan, 2018*). The cultivation and consumption of

Corresponding authors
Govindasamy Senthilraja, senthilraja@tnau.ac.in
Nagendran Tharmalingam, ntharmalingam@houstonmethodist.org

small millet are increasing in Inida. The agro-industry is now focusing on increasing small millet productivity by implementing appropriate management strategies against yield-limiting factors (*Sharma & Niranjan, 2018*).

The ascomycetous fungus *Magnaporthe grisea* (T.T. Hebert) M.E. Barr (anamorph: *Pyricularia grisea* (Cooke) Sacc.) causes blast disease in various cereals and millet worldwide (*Noman et al., 2022*; *Aravind et al., 2022*). The disease has become a significant obstacle in cultivating foxtail millet, particularly in the states of Karnataka, Tamil Nadu, Andhra Pradesh, Rajasthan, Madhya Pradesh, and Chhattisgarh in India (*Sharma et al., 2014*). The pathogen has a hemibiotrophic and heterothallic lifestyle (*Koeck, Hardham & Dodds, 2011*), which affects different parts of the plant, including leaves, leaf sheaths, nodes, necks, heads, or panicles at all stages of plant growth (*Wilson & Talbot, 2009*). Initially, the disease manifests itself as a small water-soaked lesion, after which the spots turn dark green with a central light green to greyish area. Fully developed spots have a brown to dark brown margin with a white to grey center. During the life cycle, the pathogen produces thin-walled, three-celled, pear-shaped conidia with a protruding hilum at the tips of the conidiophores, which cause and spread disease from one plant to another (*Klaubauf et al., 2014*).

Blast pathogens develop resistance to fungicides (*Suzuki et al., 2010*). Therefore, the best way to control this disease is to breed resistant cultivars. However, the pathogen can rapidly evolve new pathogenic races that overcome the plant resistance genes in the field (*Sharma et al., 2014*; *Kim et al., 2019*). Therefore, it is necessary to analyze the genetic diversity among the *Magnaporthe* populations in a particular region to understand the mechanism of the breakdown of resistance in blast-resistant cultivars (*Rieux et al., 2011*). Variation within a species can be quantified using changes in the base sequence of DNA (SNP) or the amino acid sequence of proteins, as well as a set of host differentials.

Previous studies have demonstrated genetic diversity among blast pathogens in rice (*Aravind et al., 2022*), wheat (*Noman et al., 2022*), finger millet (*Takan et al., 2012*), and pearl millet (*Sharma et al., 2021*). However, no studies have been conducted on the genetic structure of foxtail millet infecting *Magnaporthe* populations in India. Although the pathogen is variable, it is very host-specific to foxtail millet (*Sharma et al., 2014*). Improving our understanding of the genetic diversity within and among populations of blast pathogens at the field level is critical to formulating effective management strategies, including the development of blast-resistant cultivars of foxtail millet. To determine the genetic diversity of plant pathogens, microsatellites or simple sequence repeats (SSRs) are widely used marker systems (*Rieux et al., 2011*; *Li et al., 2021*). The microsatellites or SSRs are tandemly repeated DNA sequences present throughout the eukaryotic genome. Compared to other marker systems, SSRs are inexpensive (*Schoebel et al., 2013*), highly polymorphic, multi-allelic, reproducible, and user-friendly (*Nowicki et al., 2022*). Several SSR markers (*Kaye et al., 2003*) and minisatellite markers (*Li et al., 2007*) are developed already for *M. grisea*. The present study aimed to analyze the extent of genetic variability and population structure in field populations of foxtail millet infecting *M. grisea* in Tamil Nadu using SSR markers.

## MATERIALS AND METHODS

### Survey, collection, and isolation of *M. grisea*

A detailed roving survey was conducted in major foxtail millet growing areas (hills and plains) in Tamil Nadu, India during the rainy season between 2017 and 2018. Blast severity was measured on a progressive 1–9 scale, where 1 = no lesions to small brown specks of pinhead size (0.1–1.0 mm), less than 1% leaf area affected; 2 = typical blast lesions covering 1–5% leaf area covered with lesions; 3 = 6–10%, 4 = 11–20%, 5 = 21–30%, 6 = 31–40%, 7 = 41–50%, 8 = 51–75% and many leaves dead; and 9 = typical blast lesions covering >75% leaf area or all the leaves dead (*Babu et al., 2013*; *Sharma et al., 2014*) and expressed as a percent disease index (PDI) using the formula PDI = (sum of individual ratings/no. of leaves assessed × maximum disease grade value) × 100 (*Wheeler, 1969*; *Amoghavarsha et al., 2022*). Plant samples (leaves and leaf sheaths) infected with the blast were collected from the farmers' fields. From each cultivar, 25 numbers of leaves were collected. Collected samples were air-dried, bagged, labeled, and stored under refrigerated conditions at 4 °C for the isolation of pathogens. The pathogen was isolated using the standard tissue isolation procedure (*Tuite, 1969*) with potato dextrose agar (PDA) medium under aseptic conditions at 25 ± 2 °C. Mycelium growing from infected plant tissue was identified based on its morphological and cultural characteristics. The fungus was then purified by a single spore isolation technique (*Ricker & Ricker, 1936*) and the purified isolates were stored on PDA slants for further study.

### DNA extraction

Each isolate was inoculated into a 100 ml Erlenmeyer flask containing 20 ml of potato dextrose broth and incubated at 25 ± 2 °C. After 7 days of incubation, the mycelia were harvested and immediately ground in liquid nitrogen to a fine powder for DNA extraction with CTAB (Cetyltrimethylammonium bromide) buffer, following the protocol described by *Murray & Thompson (1980)* with minor modifications (*Jagadeesh et al., 2018*) using equal volumes of supernatant and a mixture of phenol-chloroform-isoamyl alcohol to precipitate and isolate the DNA from other impurities. The DNA was verified quantitatively and qualitatively using a NanoDrop 2000 UV-vis spectrophotometer (Thermo Scientific, Waltham, MA, USA) and then diluted to a working concentration of 50 ng/μl and stored at −20 °C for later use.

### PCR assays
#### ITS rDNA amplification
All the isolates were subjected to PCR amplification using a pair of universal primers, ITS1 and ITS4 (*White et al., 1990*). Primer sequences were synthesized by Eurofins Genomics India (Bangalore, India). The PCR was performed in 20 μl reaction volume with 2.0 μl template DNA (50 ng/μl), 10.0 μl master mix, 2.0 μl forward primer, 2.0 μl reverse primer, and 4.0 μl sterile double-distilled water. The reaction mixture was centrifuged briefly to thoroughly mix the components of the cocktail. Amplification was performed in a thermal cycler (C1000 Touch™ Thermal Cycler, Bio-Rad Laboratories, Inc., Singapore). Thermal cycling conditions were achieved consisting of 35 cycles, initial denaturation at 94 °C for

5 min, denaturation at 94 °C for 1min, annealing at 55 °C for 1 min, extension at 72 °C for 2 min, and final extension at 72 °C for 5 min. The PCR products were subjected to agarose (1%) gel electrophoresis in 0.5X Tris-Borate-EDTA buffer at 110V. A 100 bp DNA molecular ladder (MEDOX Biotech, Chennai, India) was used to estimate the size of the amplicon. After electrophoretic separation, the gel was read under the Gel Doc 2000 Bio-Rad system (Bio-Rad Laboratories, Hercules, CA, USA) for a more detailed analysis.

### Microsatellite genotyping

Twenty-nine microsatellite loci (SSRs) distributed across seven *M. grisea* chromosomes were used (Table S1) to analyze genetic diversity and population structure as described by *Kaye et al. (2003)* and *Adreit et al. (2007)*. These primer sequences were synthesized at Eurofins Genomics India (Bangalore, India). PCR amplification was performed in 20 μl of reaction mix in a 0.5 microcentrifuge tube using 10 ng of template DNA, 0.2 mM of dNTPs, 0.4 μM of primers, 1.5 mM MgCl2, 1X Taq buffer (10 mM Tris-HCl, 50 mM KCl, pH 8.3) and 1U Taq DNA polymerase (DreamTaq, Thermo Scientific, Waltham, MA, USA). Conditions for PCR were initial denaturation at 94 °C for 5 min, followed by 35 cycles of denaturation at 94 °C for 30 s, primer annealing at 55 °C for 45 s, and extension at 72 °C for 45 s, with the final extension for 10 min at 72 °C. For marker scoring, the PCR products were separated by electrophoresis on a 2.5% agarose gel stained with ethidium bromide. After electrophoretic separation, the gels were analyzed under the Gel Doc 2000 Bio-Rad system (Bio-Rad Laboratories, Hercules, CA, USA). All PCR reactions for each primer were repeated at least twice to confirm the data scored.

## Bioinformatics and computational and statistical analyses

The percent disease index of foxtail millet blast and observations on the conidial morphology of *M. grisea* were subjected to analyses of variance (ANOVA). The averages were compared by the Tukey test ($p < 0.05$) in the Statistical Package for the Social Sciences (SPSS), version 21.0.

The amplified PCR products were sequenced at M/S Eurofins Genomics Bangalore, India for double-pass DNA sequencing using the universal primers (ITS1 and ITS4) mentioned above. The DNA sequences were subjected to phylogenetic analysis. A neighbor-joining tree (*Saitou & Nei, 1987*) was constructed with MEGA v 6.1 software to study divergence patterns, and a 1,000-repetition bootstrap analysis was performed to support nodes in clusters (*Tamura et al., 2011*).

Basic computational and statistical analyses such as polymorphic information content (PIC), gene diversity, major allele frequency and heterozygosity were analyzed using Power Marker version 3.25 (*Liu & Muse, 2005*). Pairwise F-statistics, Nei genetic distance, analysis of molecular variance (AMOVA), and principal coordinate analysis (PCoA) were performed using the package, namely GenA1Ex v. 6.502 (*Peakall & Smouse, 2006*). PIC values measure the significance of a given DNA marker. The PIC value for each SSR locus was measured as described by *Anderson et al. (1993)*. An unweighted neighbor-joining tree was constructed based on the simple matching dissimilarity matrix of SSR markers genotyped across the *M. grisea* isolates as implemented in the DARwin v. 5.0.157 program

(*Perrier & Jacquemoud-Collet, 2006*). Population structure between *M. grisea* isolates was analyzed with the software package STRUCTURE (*Pritchard, Stephens & Donnelly, 2000*) in the revised version 2.3.4. This approach makes use of multilocus genotypes to infer the fraction of an isolate's genetic ancestry that belongs to a population for a given number of populations (K). The optimum number of populations (K) was selected after five independent runs of a burn-in of 50,000 iterations followed by 50,000 iterations for each value of K (testing from $K = 2$ to $K = 10$). The program STRUCTURE HARVESTER was used to determine the peak value of delta K according to the method described by *Evanno, Regnaut & Goudet (2005)*.

# RESULTS

## Survey and assessment of blast severity in foxtail millet

The prevalence of foxtail millet leaf blast was noticed in all surveyed areas (plains and hills) in Tamil Nadu. Interestingly, the occurrence of sheath blasts was noticed only in the Kunnur village, Salem District. The severity of leaf blast disease ranged from 10.34 to 72.19 PDI. The highest incidence was observed in the cv. CO(Te)7 (72.19 PDI) in Athiyandal village, Tiruvannamalai District, followed by a local variety from a farmer's field in Kunnur village, Salem District, which had a PDI of 70.16. The minimum PDI (10.34) was observed in the cv. CO(Te)5 in Madurai. Sampling locations were marked on the map of Tamil Nadu using ARC GIS software, and the PDI or blast severity was given in three different colors according to its range. The green color represents a lesser incidence of the disease in the Salem, Madurai, and Virudhunagar districts. The yellow color indicates a moderate disease incidence in the Thoothukudi, Dindigul, Erode, Salem, Namakkal, and Dharmapuri districts. The highest incidence of the disease was recorded in Vellore, Tiruvannamalai, Dindigul, and Virudhunagar districts and is marked in red (Fig. 1; Table 1). At the time of the survey, the leaf blast was observed by the type of lesion, which was characterized by a spindle-shaped lesion with a dark brown border with a grey center, and the sheath blast symptom was observed as a spindle-shaped lesion with a dark brown border with a white center on the sheath (Figs. 2A, 2B).

## Morphological characterization

In all, 32 isolates of *M. grisea* were isolated from the diseased samples (31 from leaf and 1 from sheath) and designated as TNFxM1 to TNFxM32 (Table 1). All isolates were classified based on morphological and conidial characteristics variations, namely colony color, surface appearance, size, shape, and color of the conidia. Colony color varied as greyish brown (15), slightly greyish brown (3), creamish white (1), white (4), greyish white (4), blackish white (3), and greyish black (2). Similarly, most of the isolates were smooth (19) and some were rough (13) in colony appearance (Fig. S1; Table 2). In all the isolates, the shape of the conidia was typically pyriform with a rounded base, narrow apex, 2 septa, 3 cells, and the middle cells were broader than the adjacent cells in all the isolates. Some of them were very long and narrow, while others were quite wide. The length and width of the conidia of different isolates of *M. grisea* were measured (Fig. S1; Table 2). The size of the conidia ranged from 20.06 μm to 37.76 μm in length and 6.88 μm to 11.93 μm in width.

**Table 1  Survey and collection of *M. grisea* isolates from different foxtail millet growing regions of Tamil Nadu.**

| S. No | Year | Isolate code | Topography | Location | District | Cultivar | PDI | Geographical information | | GenBank accession number |
|---|---|---|---|---|---|---|---|---|---|---|
| | | | | | | | | Latitude | Longitude | |
| 1. | 2017 | TNFxM1 | Plains | Athiyandal | Tiruvanamalai | CO(Te)7 | 72.19[a] | 12.13°N | 79.10°E | MN017167 |
| 2. | 2017 | TNFxM2 | Plains | K. Pudhur | Dindigul | Local | 64.65[d] | 10.15°N | 78.15°E | MT053476 |
| 3. | 2017 | TNFxM3 | Plains | Methugumalai | Dindigul | Local | 38.90[i] | 10.18°N | 78.16°E | MK990557 |
| 4. | 2017 | TNFxM4 | Plains | K. Pudhur | Dindigul | Local | 67.20[c] | 10.15°N | 78.15°E | MN028779 |
| 5. | 2017 | TNFxM5 | Plains | Seithur | Dindigul | Local | 40.19[i] | 10.22°N | 78.14°E | MN017169 |
| 6. | 2017 | TNFxM6 | Plains | Aruppukottai | Virudhunagar | CO(Te)5 | 68.90[bc] | 09.33°N | 78.50°E | MN017168 |
| 7. | 2017 | TNFxM7 | Plains | Aruppukottai | Virudhunagar | CO(Te)7 | 67.98[bc] | 09.33°N | 78.50°E | MN028778 |
| 8. | 2017 | TNFxM8 | Plains | Kovilpatti | Thoothukudi | CO(Te)7 | 35.18[j] | 09.12°N | 77.52°E | MN028776 |
| 9. | 2017 | TNFxM9 | Plains | Thummakundu | Madurai | Local | 21.35[m] | 09.53°N | 77.51°E | MT043762 |
| 10. | 2017 | TNFxM10 | Plains | Gobinathampatti | Dharmapuri | Local | 35.82[j] | 12.70°N | 78.20°E | MT043764 |
| 11. | 2017 | TNFxM11 | Hills | Pudhurnadu | Vellore | Local | 59.12[e] | 12.23°N | 78.42°E | MT043805 |
| 12. | 2017 | TNFxM12 | Plains | Madurai | Madurai | CO(Te)5 | 10.34[n] | 09.58°N | 78.12°E | MT053462 |
| 13. | 2017 | TNFxM13 | Hills | Thalavadi | Erode | Local | 46.67[h] | 11.46°N | 77.00°E | MW497608 |
| 14. | 2017 | TNFxM14 | Hills | Thalavadi | Erode | Local | 48.12[gh] | 11.47°N | 77.00°E | MW534750 |
| 15. | 2018 | TNFxM15 | Hills | Kolakoor | Salem | Local | 29.45[k] | 11.48°N | 78.12°E | MW534813 |
| 16. | 2018 | TNFxM16 | Hills | Suraikayapatti | Salem | Local | 30.65[k] | 11.52°N | 78.11°E | MW534814 |
| 17. | 2018 | TNFxM17 | Hills | Nagalur | Salem | Local | 30.98[k] | 11.51°N | 78.11°E | MW534870 |
| 18. | 2018 | TNFxM18 | Hills | Maramangalam | Salem | Local | 34.12[j] | 11.50°N | 78.18°E | MW535067 |
| 19. | 2018 | TNFxM19 | Hills | Neiyamalai | Salem | Local | 70.15[ab] | 11.78°N | 78.47°E | MW535174 |
| 20. | 2018 | TNFxM20 | Hills | Neyyamalai | Salem | Local | 69.12[bc] | 11.78°N | 78.47°E | MW494605 |
| 21. | 2018 | TNFxM21 | Hills | Neiyamalai | Salem | Local | 69.87[ab] | 11.78°N | 78.47°E | MW498280 |
| 22. | 2018 | TNFxM22 | Hills | Akkaraipati | Salem | Local | 68.13[bc] | 11.47°N | 78.06°E | MW504997 |
| 23. | 2018 | TNFxM23 | Hills | Akkaraipati | Salem | Local | 52.17[f] | 11.47°N | 78.06°E | MW535244 |
| 24. | 2018 | TNFxM24 | Plains | Valappadi | Salem | Local | 34.15[j] | 11.39°N | 78.23°E | MW504790 |
| 25. | 2018 | TNFxM25 | Plains | Valappadi | Salem | Local | 24.16[l] | 11.39°N | 78.24°E | MW535295 |
| 26. | 2018 | TNFxM26 | Hills | Karayankattupatti | Namakkal | Local | 26.15[l] | 11.16°N | 78.19°E | MW494589 |
| 27. | 2018 | TNFxM27 | Hills | Vadakkadu | Namakkal | Local | 39.34[i] | 11.23°N | 78.20°E | MW497613 |
| 28. | 2018 | TNFxM28 | Hills | Vadakkadu | Namakkal | Local | 50.12[fg] | 11.23°N | 78.20°E | MW535771 |
| 29. | 2018 | TNFxM29 | Hills | Pagadupatu | Salem | Local | 30.18[k] | 11.46°N | 78.38°E | MW504990 |
| 30. | 2018 | TNFxM30 | Hills | Adiyanur | Salem | Local | 69.67[b] | 11.42°N | 78.37°E | MW494316 |
| 31. | 2018 | TNFxM31 | Hills | Kunnur (leaf) | Salem | Local | 70.16[ab] | 11.43°N | 78.37°E | MW535961 |
| 32. | 2018 | TNFxM32 | Hills | Kunnur (sheath) | Salem | Local | 11.24[n] | 11.43°N | 78.37°E | MW496127 |

**Notes.**

Means followed by the same letters were not significantly different from each other according to Tukey's test ($P < 0.05$).

PDI, Percent disease index.

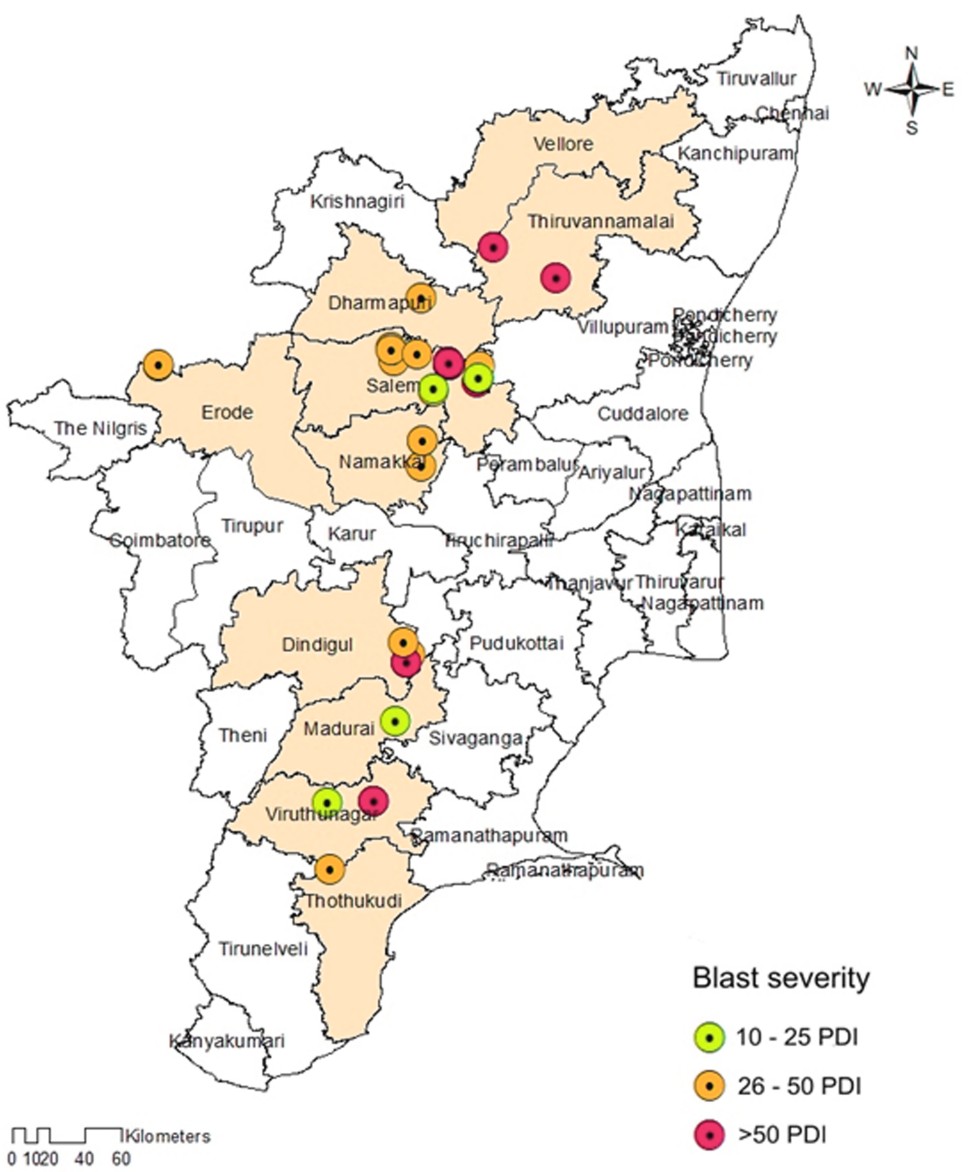

**Figure 1** **Map showing the collection sites of *Magnaporthe grisea* isolates and foxtail millet blast severity in Tamil Nadu.** Sampling locations marked on the map of Tamil Nadu were extracted using ArcGis software. Map source: ArcGis.

All isolates varied significantly in terms of conidial size. The blast isolates TNFxM7 and TNFxM20 had the longest conidia of 37.76 μm and 33.83 μm, respectively. The shortest conidial length was observed in isolates TNFxM23 (20.06 μm) and TNFxM6 (21.48 μm). The highest conidial width was observed in isolates TNFxM26 and TNFxM27 at 11.93 μm and 10.77 μm, respectively.

(a)           (b)

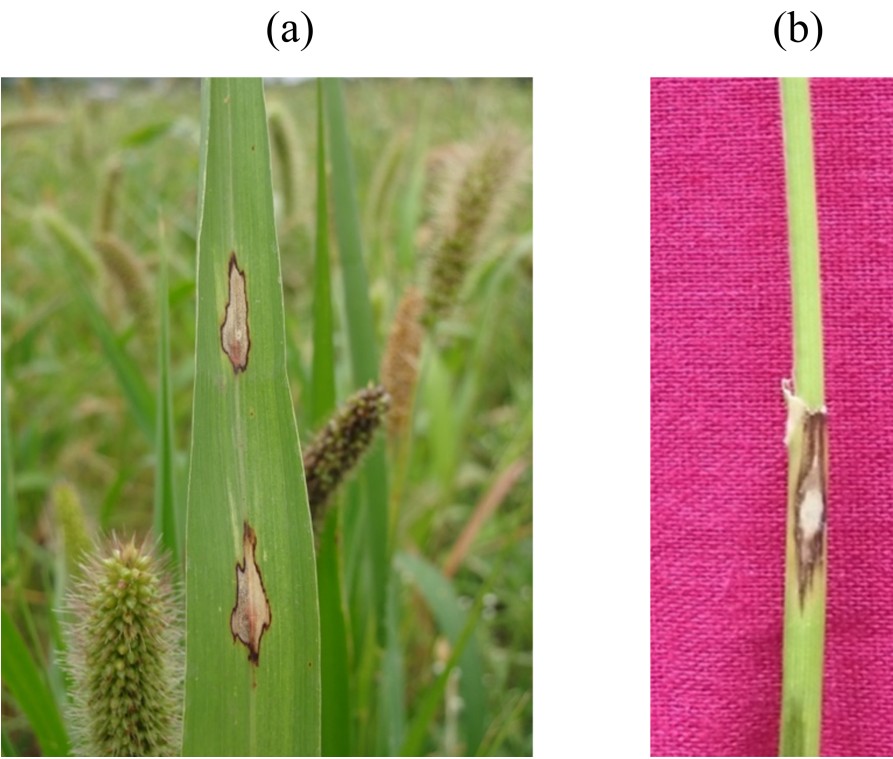

**Figure 2   Foxtail millet blast symptoms.** (A) Leaf blast: spindle shaped lesion having dark brown margin with grey colored center, (B) Sheath blast: spindle shaped lesion having dark brown margin with white colored center.

### ITS-DNA barcoding and phylogenetic analysis

ITS-based primers amplified the expected amplicon size of 560 bp (Fig. S2). All the sequences obtained from this study by partial gene sequencing showed more than 99% sequence similarity to GenBank sequences (*M. grisea*) deposited with the NCBI. Accession numbers obtained from the NCBI GenBank for each isolate are listed in Table 1.

The phylogenetic analysis revealed that all 32 *M. grisea* isolates were divided into two groups (Group I and Group II). The two isolates (MN017168 and MN028778) from Virudhunagar District were only clustered separately and formed group I. Group II consists of thirty isolates with a further subdivision into subgroups I and subgroup II. Subgroup I was formed with eight isolates from Tiruvanamalai (MN017167), Salem (MW534814, MW535244, MW504790, MW535295, MW535961 and MW496127) and Namakkal (MW497613) districts. The remaining twenty-two isolates from Dindigul (MT053476, MK990557, MN028779, and MN017169), Thoothukudi (MN028776), Madurai (MT043762 and MT053462), Dharmapuri (MT043764), Vellore (MT043805), Erode (MW497608 and MW534750), Salem (MW534813, MW534870, MW535067, MW535174, MW494605, MW498280, MW504997, MW504990, and MW494316) and Namakkal (MW494589 and MW535771) districts were grouped altogether and formed subgroup II (Fig. 3).

**Table 2 Morphological and conidial characteristics of *M. grisea* isolates.**

| S. No | Isolates | Color of vegetative growth | Colony texture | Conidial size (μm) | | Conidial shape | Conidial color |
|---|---|---|---|---|---|---|---|
| | | | | Length | Width | | |
| 1. | TNFxM1 | Slight greyish brown | Rough surface | 22.64$^l$ | 9.12$^{fg}$ | Pyriform | Hyaline to pale olive |
| 2. | TNFxM2 | Slight greyish brown | Smooth surface | 22.80$^l$ | 6.88$^{mn}$ | Pyriform | Hyaline to pale olive |
| 3. | TNFxM3 | Creamish white | Smooth surface | 32.17$^c$ | 11.89$^a$ | Pyriform | Hyaline to pale olive |
| 4. | TNFxM4 | Slight greyish brown | Smooth surface | 25.63$^{jk}$ | 7.72$^{ij}$ | Pyriform | Hyaline to pale olive |
| 5. | TNFxM5 | Greyish brown | Rough surface | 26.84$^{ij}$ | 10.37$^c$ | Pyriform | Hyaline to pale olive |
| 6. | TNFxM6 | Greyish brown | Rough surface | 21.48$^l$ | 7.51$^{jk}$ | Pyriform | Hyaline to pale olive |
| 7. | TNFxM7 | Greyish brown | Rough surface | 37.76$^a$ | 7.12$^{mn}$ | Pyriform | Hyaline to pale olive |
| 8. | TNFxM8 | Greyish brown | Rough surface | 24.32$^k$ | 8.56$^h$ | Pyriform | Hyaline to pale olive |
| 9. | TNFxM9 | Greyish brown | Rough surface | 30.00$^{de}$ | 8.98$^g$ | Pyriform | Hyaline to pale olive |
| 10. | TNFxM10 | Greyish brown | Rough surface | 25.93$^j$ | 9.30$^{ef}$ | Pyriform | Hyaline to pale olive |
| 11. | TNFxM11 | White | Smooth surface | 27.44$^{ghi}$ | 9.68$^{de}$ | Pyriform | Hyaline to pale olive |
| 12. | TNFxM12 | Greyish brown | Smooth surface | 26.69$^{ij}$ | 7.94$^i$ | Pyriform | Hyaline to pale olive |
| 13. | TNFxM13 | Greyish white | Smooth surface | 28.63$^{fg}$ | 8.80$^{gh}$ | Pyriform | Hyaline to pale olive |
| 14. | TNFxM14 | Greyish white | Smooth surface | 28.41$^{fgh}$ | 8.43$^h$ | Pyriform | Hyaline to pale olive |
| 15. | TNFxM15 | Greyish white | Smooth surface | 28.52$^{fg}$ | 8.41$^h$ | Pyriform | Hyaline to pale olive |
| 16. | TNFxM16 | White | Smooth surface | 27.30$^{hi}$ | 8.43$^h$ | Pyriform | hyaline to pale olive |
| 17. | TNFxM17 | Greyish brown | Rough surface | 27.26$^{hi}$ | 7.47$^{jk}$ | Pyriform | Hyaline to pale olive |
| 18. | TNFxM18 | Blackish white | Rough surface | 26.77$^{ij}$ | 9.02$^{fg}$ | Pyriform | Hyaline to pale olive |
| 19. | TNFxM19 | Greyish brown | Smooth surface | 30.80$^d$ | 7.20$^{km}$ | Pyriform | Hyaline to pale olive |
| 20. | TNFxM20 | Greyish white | Smooth surface | 33.88$^b$ | 10.56$^{bc}$ | Pyriform | Hyaline to pale olive |
| 21. | TNFxM21 | Greyish brown | Smooth surface | 30.61$^{de}$ | 6.80$^n$ | Pyriform | Hyaline to pale olive |
| 22. | TNFxM22 | Greyish brown | Smooth surface | 24.83$^k$ | 9.87$^d$ | Pyriform | Hyaline to pale olive |
| 23. | TNFxM23 | Greyish black | Smooth surface | 20.06$^m$ | 9.60$^d$ | Pyriform | Hyaline to pale olive |
| 24. | TNFxM24 | Greyish brown | Rough surface | 27.12$^{ghij}$ | 9.06$^{fg}$ | Pyriform | Hyaline to pale olive |
| 25. | TNFxM25 | Greyish brown | Rough surface | 26.43$^{ij}$ | 9.80$^d$ | Pyriform | Hyaline to pale olive |
| 26. | TNFxM26 | White | Smooth surface | 25.68$^{jk}$ | 11.93$^a$ | Pyriform | Hyaline to pale olive |
| 27. | TNFxM27 | Blackish white | Smooth surface | 27.90$^{fghi}$ | 10.77$^b$ | Pyriform | Hyaline to pale olive |
| 28. | TNFxM28 | Blackish white | Rough surface | 28.44$^{fgh}$ | 9.67$^{de}$ | Pyriform | Hyaline to pale olive |
| 29. | TNFxM29 | Greyish black | Rough surface | 29.35$^{ef}$ | 8.98$^{fg}$ | Pyriform | Hyaline to pale olive |
| 30. | TNFxM30 | White | Smooth surface | 27.44$^{ghi}$ | 8.56$^h$ | Pyriform | Hyaline to pale olive |
| 31. | TNFxM31 | Greyish brown | Smooth surface | 28.41$^{fgh}$ | 8.43$^h$ | Pyriform | Hyaline to pale olive |
| 32. | TNFxM32 | Greyish brown | Smooth surface | 26.94$^{hij}$ | 7.94$^i$ | Pyriform | Hyaline to pale olive |

**Notes.**
Values are mean of two observations. Means followed by the same letters were not significantly different from each other according to Tukey's test ($P < 0.05$).

## SSR genotyping and cluster analysis

All 32 isolates were genotyped with 29 SSR markers distributed throughout the *M. grisea* genome. A total of 103 alleles were detected among the 29 SSR markers, ranging from two (Pyrms 67–68, Pyrms 107–108, and Pyrms 533–534) to six (Pyrms 607–608) alleles with an average of 3.55 alleles per locus. The major allele frequency ranged from 0.344 (Pyrms 83–84) to 0.906 (Pyrms 533–534), with a mean of 0.605. Genetic diversity ranged from 0.170 (Pyrms 533–534) to 0.717 (Pyrms 657–658) with a mean value of 0.517. The

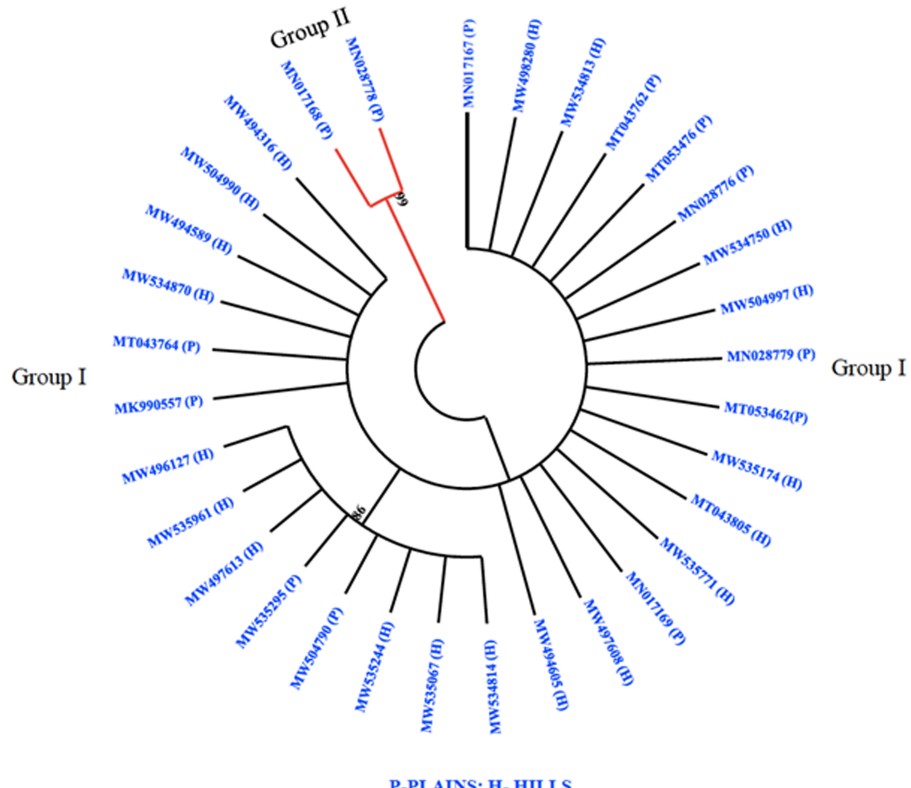

**Figure 3 Phylogenetic analysis.** Phylogenetic tree constructed based on multiple alignments of nucleotide sequences of *M. grisea* isolates. The tree was generated using the neighbor joining method with 1,000 bootstrap replications and the tree drawn at a cut-off value of 60% in MEGA 6.1.

polymorphism information content (PIC) of the markers had an average value of 0.465 and varied from 0.155 to 0.680. The maximum value of PIC was observed for marker Pyrms 37–38, while the minimum value was observed for marker Pyrms 533–534. The observed mean heterozygosity was 0.007517 with a range of 0.00 to 0.094. Among the markers used, only 5 (Pyrms 37–38, Pyrms 39–40, Pyrms 99–100, Pyrms 427–428, and Pyrms 453–454) showed a heterozygosity value greater than zero, while 24 had zero values (Table 3). The unrooted unweighted neighbor Joining method was used to estimate the genetic distance and dissimilarity index of 32 *M. grisea* isolates. For the unrooted tree, isolates were classified into three major clusters (Fig. 4). Among the three clusters, cluster 1 (red color) recorded 14 isolates, while cluster 2 (blue color) consisted of 16 isolates, and cluster 3 (green color) had 2 isolates.

## Population structure analysis

For genetic structure estimation, a Bayesian clustering approach was followed by taking probable subpopulations (K), and a higher delta K value using STRUCTURE 2.3.6 software. The maximum plateau of *adhoc* measurement of $\Delta K$ was $K = 2$ with a $\Delta K$ value of 219.8 (Table S2). The population structure was analyzed for genetic relatedness among 32 *M. grisea* isolates collected from ten different districts of Tamil Nadu, India.

**Table 3  Genetic analysis of *M. grisea* isolates with 29 SSR markers.**

| S. No. | Marker | Major allele frequency | Number of alleles | Gene diversity | Heterozygosity | PIC |
|---|---|---|---|---|---|---|
| 1. | Pyrms 7–8 | 0.613 | 3.000 | 0.547 | 0.000 | 0.487 |
| 2. | Pyrms 15–16 | 0.500 | 4.000 | 0.633 | 0.000 | 0.570 |
| 3. | Pyrms 37–38 | 0.453 | 5.000 | 0.716 | 0.031 | 0.680 |
| 4. | Pyrms 39–40 | 0.453 | 3.000 | 0.639 | 0.031 | 0.565 |
| 5. | Pyrms 41–42 | 0.438 | 4.000 | 0.643 | 0.000 | 0.572 |
| 6. | Pyrms 43–44 | 0.688 | 4.000 | 0.484 | 0.000 | 0.443 |
| 7. | Pyrms 45–46 | 0.500 | 4.000 | 0.615 | 0.000 | 0.544 |
| 8. | Pyrms 47–48 | 0.813 | 4.000 | 0.328 | 0.000 | 0.313 |
| 9. | Pyrms 59–60 | 0.844 | 3.000 | 0.275 | 0.000 | 0.257 |
| 10. | Pyrms 61–62 | 0.438 | 4.000 | 0.650 | 0.000 | 0.585 |
| 11. | Pyrms 63–64 | 0.656 | 3.000 | 0.498 | 0.000 | 0.436 |
| 12. | Pyrms 67–68 | 0.875 | 2.000 | 0.219 | 0.000 | 0.195 |
| 13. | Pyrms 77–78 | 0.563 | 3.000 | 0.588 | 0.000 | 0.523 |
| 14. | Pyrms 81–82 | 0.438 | 3.000 | 0.619 | 0.000 | 0.539 |
| 15. | Pyrms 83–84 | 0.344 | 4.000 | 0.707 | 0.000 | 0.651 |
| 16. | Pyrms 87–88 | 0.500 | 4.000 | 0.635 | 0.000 | 0.574 |
| 17. | Pyrms 93–94 | 0.469 | 3.000 | 0.635 | 0.000 | 0.561 |
| 18. | Pyrms 99–100 | 0.453 | 3.000 | 0.625 | 0.031 | 0.546 |
| 19. | Pyrms 101–102 | 0.844 | 3.000 | 0.275 | 0.000 | 0.257 |
| 20. | Pyrms 107–108 | 0.875 | 2.000 | 0.219 | 0.000 | 0.195 |
| 21. | Pyrms 109–110 | 0.594 | 3.000 | 0.564 | 0.000 | 0.503 |
| 22. | Pyrms 125–126 | 0.438 | 4.000 | 0.627 | 0.000 | 0.552 |
| 23. | Pyrms 233–234 | 0.516 | 4.000 | 0.631 | 0.000 | 0.570 |
| 24. | Pyrms 319–320 | 0.656 | 4.000 | 0.521 | 0.000 | 0.479 |
| 25. | Pyrms 427–428 | 0.859 | 4.000 | 0.254 | 0.031 | 0.244 |
| 26. | Pyrms 453–454 | 0.594 | 5.000 | 0.602 | 0.094 | 0.569 |
| 27. | Pyrms 533–534 | 0.906 | 2.000 | 0.170 | 0.000 | 0.155 |
| 28. | Pyrms 607–608 | 0.844 | 3.000 | 0.275 | 0.000 | 0.257 |
| 29. | Pyrms 657–658 | 0.406 | 6.000 | 0.717 | 0.000 | 0.673 |
| | Mean | 0.605862 | 3.55 | 0.514172 | 0.007517 | 0.465345 |

Based on an ancestry threshold of >70%, all 32 isolates were classified into two major populations and considered pure, while <70% were considered admixture (AD). Among the 32 isolates, 13 isolates collected from Salem and Namakkal districts were classified as subpopulation 1 (SP1), and 16 isolates collected from Tiruvanamalai, Dindigul, Virudhunagar, Thoothukudi, Madurai, Dharmapuri, Vellore, Madurai, Erode and Salem districts were classified as subpopulation 2 (SP2), while three isolates (TNFxM6, TNFxM17, and TNFxM18) were of admixture type with a major genetic component of two subgroups (Table 4). The population shows admixture, suggesting that there is a gene flow between these populations. The fixation index ($F_{st}$) values of the two populations were 0.4296 for SP1 and 0.2021 for SP2. The highest divergence in allele frequency between populations

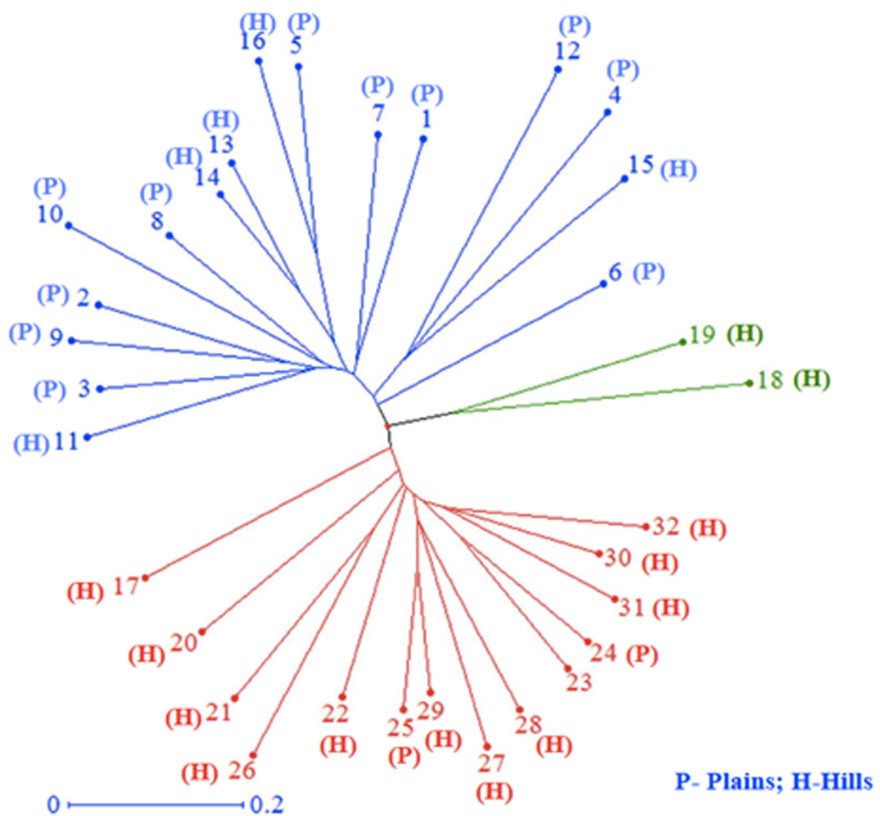

**Figure 4 Cluster analysis.** Neighbor–joining cluster analysis of 32 blast isolates based on 29 SSR markers using DARwin software.

was observed in SP1 and SP2 (0.1077) based on the net nucleotide distance calculated using point estimates of P. The mean distance (expected heterozygosity) between individuals in the panel population was 0.2036 and 0.2800 in response to SP1 and SP2, respectively (Figs. 5A, 5B; Figs. S3A–S3C).

## Analysis of molecular variance (AMOVA)

The two populations, *i.e.,* subpopulation 1 (13), and subpopulation 2 (16), along with the admixtures (3) generated from structure analysis, were analyzed to determine genetic variation among and within populations. In the AMOVA analysis, maximum variation (79%) was found among individuals, while minimum variation (20%) was noticed between the populations and within individuals (1%) (Fig. 6; Table S3). The deviation from Hardy-Weinberg's prediction was performed using Wright's F-statistics. The $F_{is}$ and $F_{it}$ values for all 29 loci were found to be 0.984 and 0.987 (r = <0.001), respectively, while $F_{st}$ between populations was 0.202. The NM value of assumed subpopulations was noted as 0.989. The highest pairwise genetic Nei distance was observed between SP1 and AD (0.404), followed by SP1 and SP2 (0.362) and SP2 and AD (0.307). Furthermore, the biplot generated from the principal coordinate analysis (PCoA) showed that the first two components account

**Table 4  Population structure group of *M. grisea* based on inferred ancestry values.**

| S. No | Isolates | Location (districts) | Topography | Inferred ancestry at $K = 2$ | | Sub population |
|---|---|---|---|---|---|---|
| | | | | Q1 | Q2 | |
| 1. | TNFxM1 | Tiruvanamalai | Plains | 0.062 | 0.938 | SP2 |
| 2. | TNFxM2 | Dindigul | Plains | 0.063 | 0.937 | SP2 |
| 3. | TNFxM3 | Dindigul | Plains | 0.048 | 0.952 | SP2 |
| 4. | TNFxM4 | Dindigul | Plains | 0.015 | 0.985 | SP2 |
| 5. | TNFxM5 | Dindigul | Plains | 0.004 | 0.996 | SP2 |
| 6. | TNFxM6 | Virudhunagar | Plains | 0.311 | 0.689 | AD |
| 7. | TNFxM7 | Virudhunagar | Plains | 0.144 | 0.856 | SP2 |
| 8. | TNFxM8 | Thoothukudi | Plains | 0.060 | 0.940 | SP2 |
| 9. | TNFxM9 | Madurai | Plains | 0.009 | 0.991 | SP2 |
| 10. | TNFxM10 | Dharmapuri | Plains | 0.004 | 0.996 | SP2 |
| 11. | TNFxM11 | Vellore | Hills | 0.017 | 0.983 | SP2 |
| 12. | TNFxM12 | Madurai | Plains | 0.252 | 0.748 | SP2 |
| 13. | TNFxM13 | Erode | Hills | 0.005 | 0.995 | SP2 |
| 14. | TNFxM14 | Erode | Hills | 0.007 | 0.993 | SP2 |
| 15. | TNFxM15 | Salem | Hills | 0.016 | 0.984 | SP2 |
| 16. | TNFxM16 | Salem | Hills | 0.034 | 0.966 | SP2 |
| 17. | TNFxM17 | Salem | Hills | 0.600 | 0.400 | AD |
| 18. | TNFxM18 | Salem | Hills | 0.374 | 0.626 | AD |
| 19. | TNFxM19 | Salem | Hills | 0.288 | 0.712 | SP2 |
| 20. | TNFxM20 | Salem | Hills | 0.808 | 0.192 | SP1 |
| 21. | TNFxM21 | Salem | Hills | 0.977 | 0.023 | SP1 |
| 22. | TNFxM22 | Salem | Hills | 0.981 | 0.019 | SP1 |
| 23. | TNFxM23 | Salem | Hills | 0.980 | 0.020 | SP1 |
| 24. | TNFxM24 | Salem | Plains | 0.995 | 0.005 | SP1 |
| 25. | TNFxM25 | Salem | Plains | 0.993 | 0.007 | SP1 |
| 26. | TNFxM26 | Namakkal | Hills | 0.897 | 0.103 | SP1 |
| 27. | TNFxM27 | Namakkal | Hills | 0.995 | 0.005 | SP1 |
| 28. | TNFxM28 | Namakkal | Hills | 0.996 | 0.004 | SP1 |
| 29. | TNFxM29 | Salem | Hills | 0.992 | 0.008 | SP1 |
| 30. | TNFxM30 | Salem | Hills | 0.995 | 0.005 | SP1 |
| 31. | TNFxM31 | Salem | Hills | 0.994 | 0.006 | SP1 |
| 32. | TNFxM32 | Salem | Hills | 0.982 | 0.018 | SP1 |

for 100% of the variations (Figs. 7A, 7B). The scatterplot was developed from the PCoA analysis and displayed across isolates on the first two axes. The PCoA 1 biplot captured 18.03% of the variation, while PCoA 2 contributed 9.13% to the total genetic variation, suggesting that the total genetic variation among the populations was 27.16% (Tables S4A, S4B).

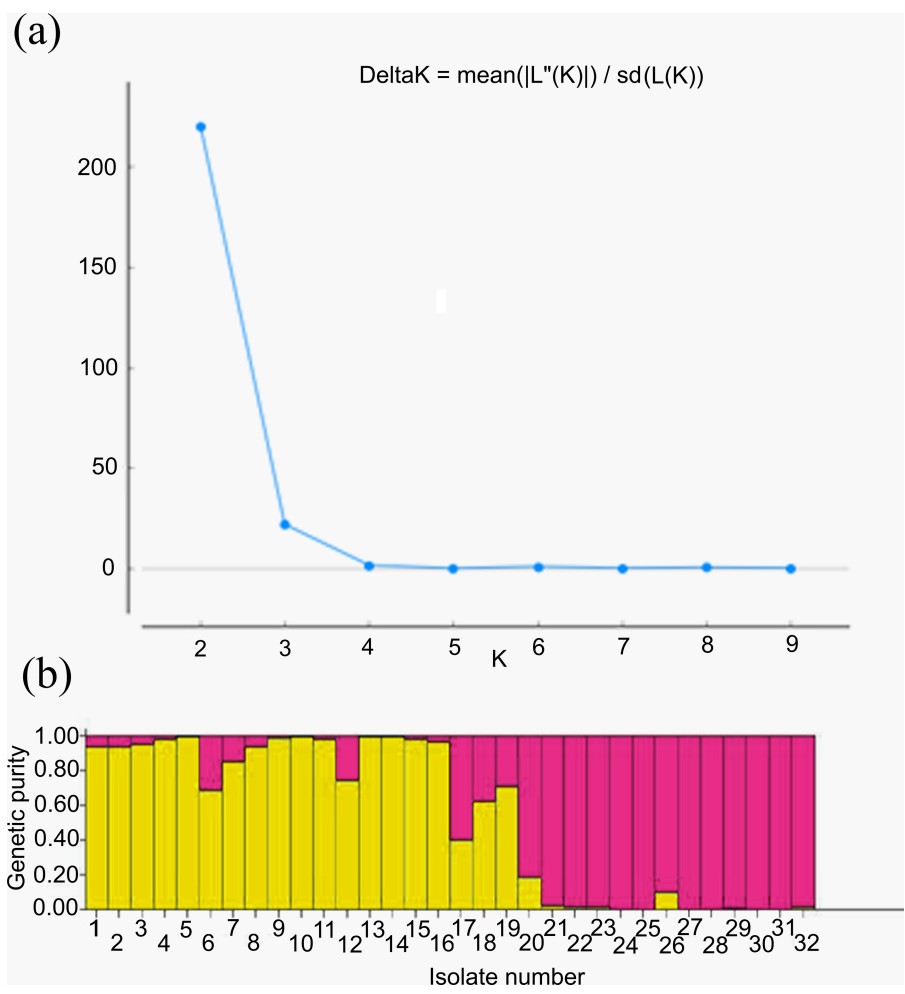

**Figure 5 Population structure analysis.** (A) Values of DK, with its modal value used to detect true K of the group ($K = 2$). For each K value, at least three independent runs were considered and averaged over the replicates, (B) Population structure of 32 blast isolates based on 29 markers ($K = 2$) and graph of estimated membership fraction for $K = 2$. The maximum of *adhoc* measure Δ K determined by structure harvester was found to be $K = 2$, which indicated that the entire population can be grouped into two subgroups. Different color within group indicates the proportion of shared ancestry with other group which has the same color with the admixture.

## DISCUSSION

Foxtail millet is grown in both the plain and hilly regions of Tamil Nadu in India. The blast disease caused by *M. grisea* (*Sharma et al., 2014*), caused a vast yield loss. We examined the morphological and genetic variability of 32 *M. grisea* isolates using 29 microsatellite markers (*Kaye et al., 2003*; *Adreit et al., 2007*) and universal ITS primers (ITS1 and ITS4) to characterize the genetic diversity and population structure of foxtail millet infecting *M. grisea*. We demonstrated the genetic analyses of foxtail millet blast populations, which can be used to develop blast-resistant foxtail millet cultivars.

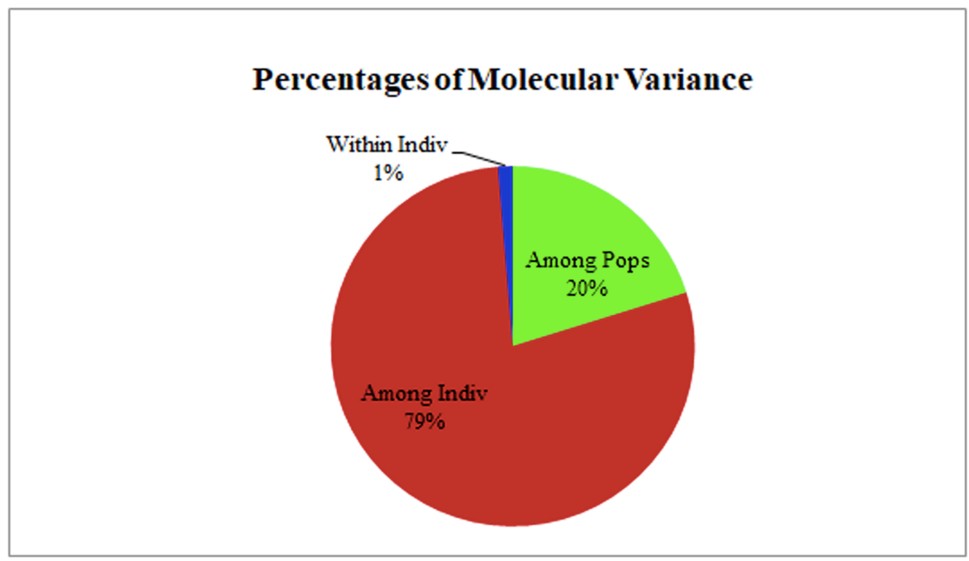

**Figure 6** AMOVA analysis of foxtail millet infecting *M. grisea* isolates.

In the present study, the severity of the blast was assessed in various foxtail millet growing areas in Tamil Nadu during the rainy season between 2017 and 2018. The highest incidence of the disease was observed in the Vellore, Tiruvannamalai, Dharmapuri, and Dindigul districts of Tamil Nadu. Similarly, *Sharma et al. (2014)* also noticed the occurrence of blast incidence during the rainy season in 2008 in different foxtail millet growing tracts in Telangana and Andhra Pradesh. We isolated the causative agent of foxtail millet blast from the infected leaf and sheath samples collected at surveyed fields using a PDA medium and purified by the single spore isolation technique. Other reports demonstrating that a similar single-spore isolation technique was used to isolate and purify *Magnaporthe* from rice, wheat, finger millet, and foxtail millet using different media (*Sharma et al., 2014*; *Yadav et al., 2019*) were discussed and corroborated by our study.

The present study demonstrates a variation in mycelium color and texture variation among the 32 fungal isolates collected in Tamil Nadu. This confirms previous reports by *Panda et al. (2017)* and *Sahu et al. (2018)*, who classified the rice blast isolates collected from different parts of Chhattisgarh and Odisha in India into different groups based on colony color and texture. The present study shows the variation among the 32 isolates of *M. grisea* in terms of morphological and conidial characteristics. However, there is no significant difference between the colony morphology and conidial characters concerning the collection period and geographical distribution of the isolates.

Sequencing of the ITS region of rDNA has become the universal barcode system for identifying fungal taxonomy at the species level (*Paloi et al., 2022*). In the present study, all *M. grisea* isolates were amplified using universal primers (ITS1 and ITS4) and the result showed the expected 560 bp amplicon in all isolates. The isolates showed a sequence homology of >99% with *M. grisea*. The results of the present study are consistent with *Jagadeesh et al. (2018)* who examined the diversity of 72 isolates of *M. oryzae* collected in

(a)

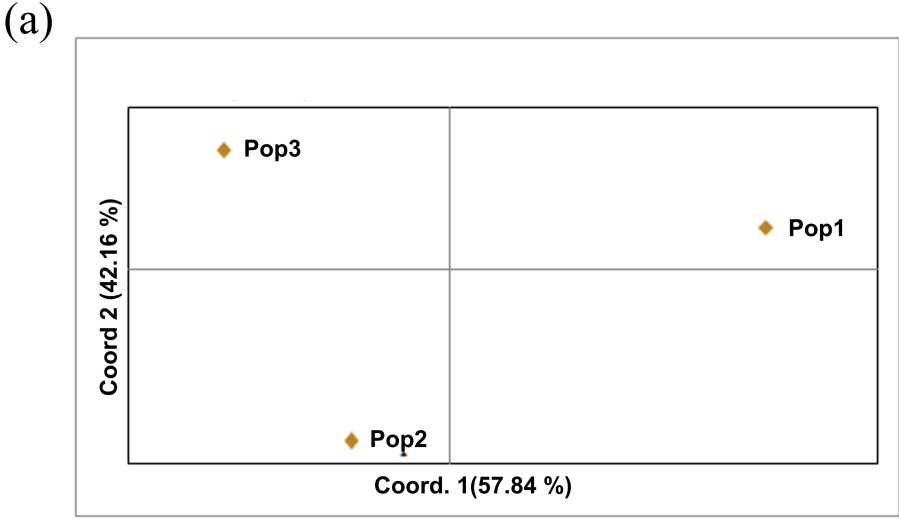

(b)

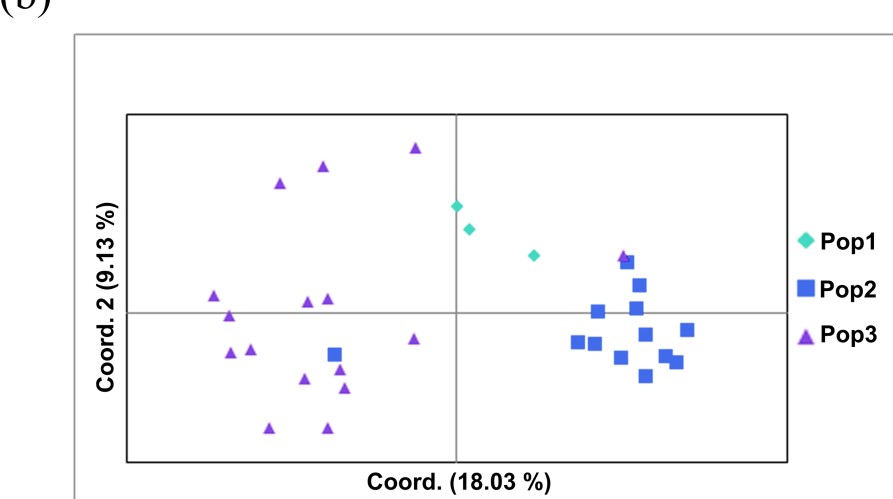

**Figure 7 Principal coordinate analysis (PCoA).** (A) PCoA for Nei genetic diversity among the three population, (B) PCoA of 32 blast isolates in the panel population using 29 molecular markers. The isolates are colored on the basis of sub-populations obtained from structure analysis (SP1-yellow, SP2-green, AD-red).

Karnataka, India using primers ITS1 and ITS4. The two clades shown in the phylogenetic analysis in the present work demonstrate the degree of diversity among the 32 isolates of *M. grisea* isolated from different districts of Tamil Nadu in India, again reflecting the results of *Jagadeesh et al. (2018)*, where genetic variations were observed in 72 isolates of the rice blast pathogen in the ITS region. The current study identified taxonomic relationships or differences between 32 isolates of foxtail millet *M. grisea*, which may be useful for identifying the resistance or susceptibility to a particular genotype. Phylogenetic analysis based on ITS region sequences revealed that all the thirty-two *M. grisea* isolates could be divided into two groups (Group I and Group II). The two isolates (MN017168 and MN028778) collected from the Virudhunagar district formed separate clusters as Group

I and the remaining thirty isolates formed Group II. Lineages are most often split by the migration of a few individuals to a newly isolated region. The two Virudhunagar isolates (where we had not previously observed a foxtail millet blast) formed separate clusters as Group I, possibly due to geological or climatic events. As a result of this genetic isolation, the lineages will evolve separately and become increasingly distinct in their physiological and behavioral patterns over time.

Microsatellites or SSRs are powerful markers used for population genetic analyses due to their high specificity, polymorphism and reproducibility (*Schoebel et al., 2013*; *Nowicki et al., 2022*). We evaluated 29 SSR markers reported by *Kaye et al. (2003)* and *Adreit et al. (2007)* to analyze genetic diversity in the population of foxtail millet infecting *M. grisea* from Tamil Nadu. To the best of our knowledge, this is the first study to investigate the molecular diversity and population structure of 32 isolates of foxtail millet infecting *M. grisea* from Tamil Nadu using microsatellites. The SSR markers used in this study varied widely and the results were generally consistent with previous population genetic studies conducted in India and elsewhere (*Yadav et al., 2019*). In the present study, 103 alleles were detected using 29 microsatellite markers. The number of alleles per locus ranged from 2 to 6, averaging 3.55 alleles/locus. A variation in the number of alleles was observed between the markers used. Previous studies reported similar results in rice (*Aravind et al., 2022*), wheat (*Noman et al., 2022*), finger millet (*Takan et al., 2012*), and pearl millet (*Sharma et al., 2021*) infecting populations of *Magnaporthe* in different geographic locations. Some previous studies showed that the high number of alleles per locus resulted in high genetic variation. The high genetic diversity was demonstrated in the present study, confirming previous results of *Saleh et al. (2014)*, while *Yadav et al. (2019)* reported a low level of genetic diversity among the *M. oryzae* isolates collected in North Eastern India.

Similarly, high levels of polymorphism were found among the 29 microsatellites used in this study. The results agree with *Wang et al. (2017)*, where a high degree of polymorphism was observed when analyzing 457 rice blast isolates collected in the United States using ten SSR markers. Recently, *Adhikari et al. (2020)* also observed high levels of polymorphism when analyzing 17 pearl millet blast isolates collected in India using SSR markers. However, a low level of polymorphism has been observed in the rice blast population in Brazil and Japan (*Prabhu, Filippi & Araujo, 2002*). Analysis of genetic diversity showed a high percentage of identical alleles, indicating considerable gene flow between all isolates of the *M. grisea* population in Tamil Nadu. Gene flow is very essential in agroecosystems for plant pathogens since it is the process by which new genes are transferred into agricultural fields far from the original mutation location. This technique is likely to play a role in natural ecosystems as well in the context of changing environmental conditions.

As most foxtail millet cultivation in Tamil Nadu is dominated by a single cultivar CO(Te)7 and no resistant cultivar is grown yet, there is no selection pressure on the pathogen to induce change. Hence this could be why there is a high percentage of identical alleles between populations of *M. grisea*. The dispersal of *M. grisea* from one location to another through airborne conidia and contaminated seeds was considered an important factor in the introduction of *M. grisea* to other parts of India. This is also reflected in the extent of migration between the plains and the hills. The hilly region seems to have a higher

migration rate than the plains. This is probably due to the different weather patterns in the hills (cold) and the plains (warm). In general, the results indicate the possibility of free movement of the pathogen between the regions. The present study undoubtedly shows an excellent diversity of *M. grisea* in the hilly region and that most of the variability was within individuals. Our results agree with those of *Wang et al. (2017)*, who analyzed 457 rice blast isolates using the neighbor-joining method to build an unrooted tree. A total of five major clusters were identified. The observed clusters were mainly related to the sampling period but not to the geographic location of the isolates.

A Bayesian clustering approach was used to analyze the structure of the foxtail millet blast isolates. The structure analysis divided the *M. grisea* isolates into two subpopulations ($K = 2$) with three admixtures. The first subpopulation (SP1) consisted of isolates from the Salem and Nammakal districts of Tamil Nadu. Similarly, the second subpopulation (SP2) consisted of Tiruvannamalai, Madurai, Dindigul, Erode, Virudhunagar, Tuticorin, Dharmapuri, Vellore, and Salem districts in Tamil Nadu. Admixtures (AD) present in isolates can increase genetic diversity. However, the structure analysis could not differentiate the blast isolates based on the geographic location. Similarly, *Yadav et al. (2019)* identified two subpopulations of leaf and neck blast isolates infecting rice crops and showed limited categorization of blast isolates based on the location. Interestingly, *Onaga et al. (2015)* categorized the 88 rice blast isolates collected from East Africa into five ancestral genetic clusters ($K = 5$) with Structure V 2.3. Package. *Wang et al. (2017)* analyzed 457 blast isolates using ten polymorphic microsatellites. They identified six genetic clusters ($K = 6$) based on collection period but not geographic location. However, a PCoA analysis was performed from the results obtained from interpreting the structure. The AMOVA analysis was worked out by dividing the genetic diversity within and between the populations of 32 isolates of foxtail millet blast. Variation between these populations contributed 20% of the genetic diversity, while 79% of the genetic variation was present between the individuals; however, one percent of the variation was found within individuals. A higher genetic diversity was found among the blast isolates than in the population. Identically, analysis of molecular variance values for genetic diversity in the pathogen population varied from 78.66% (*Wang et al., 2017*) to 86.6% (*D'Ávila et al., 2016*), 88.09% (*Onaga et al., 2015*) and 98% (*Yadav et al., 2019*). Our results are consistent with previous reports in which the pathogen population was not differentiated by geographic location (*Onaga et al., 2015*).

The genetic variation and population structure of *M. grisea* infecting foxtail millet from different regions of Tamil Nadu shows significant genetic variation within populations, similar to those observed in other population studies of *M. grisea* in India and elsewhere. Our results showed that gene flow occurs between regions and has significant implications for foxtail millet growers when virulent or fungicide-resistant strains move between regions. Preventing such a threat from the foxtail millet blast requires proper prediction and forecasting systems. In addition, long-term monitoring is essential to identify the population origin and evolutionary potential of foxtail millet blast isolates in India.

### Funding

The research project was funded by the Science and Engineering Research Board (SERB), Department of Science and Technology (DST), New Delhi, India (ECR/2016/000982). P20GM121344 from the National Institutes of Health and the National Institute of General Medical Sciences award to COBRE- CARTD (Center for Antimicrobial Resistance and Therapeutic Discovery) supports Nagendran Tharmalingam *via* a pilot grant. The funders had no role in study design, data collection and analysis, decision to publish, or preparation of the manuscript.

### Grant Disclosures

The following grant information was disclosed by the authors:
Science and Engineering Research Board (SERB), Department of Science and Technology (DST), New Delhi, India: ECR/2016/000982.
National Institutes of Health and the National Institute of General Medical Sciences: P20GM121344.

### Competing Interests

Nagendran Tharmalingam is an Academic Editor of PeerJ. The authors declare that they have no conflicts of interest.

### Author Contributions

- Manimozhi Dhivya conceived and designed the experiments, performed the experiments, analyzed the data, prepared figures and/or tables, authored or reviewed drafts of the article, and approved the final draft.
- Govindasamy Senthilraja acquired the fund and administered the project, conceived and designed the experiments, performed the experiments, analyzed the data, prepared figures and/or tables, authored or reviewed drafts of the article, and approved the final draft.
- Nagendran Tharmalingam analyzed the data, prepared figures and/or tables, editing, analysis tools, and approved the final draft.
- Sankarasubramanian Harish analyzed the data, prepared figures and/or tables, and approved the final draft.
- Kalaiselvan Saravanakumari analyzed the data, prepared figures and/or tables, and approved the final draft.
- Theerthagiri Anand analyzed the data, prepared figures and/or tables, and approved the final draft.
- Sundararajan Thiruvudainambi analyzed the data, prepared figures and/or tables, and approved the final draft.

### Data Availability

The raw data related to the genetic analysis of M. grisea isolates using 29 SSR markers is available in the Supplementary File.

## Supplemental Information

Supplemental information for this article can be found online at http://dx.doi.org/10.7717/peerj.16258#supplemental-information.

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
