# Peer review of "Analysis of genetic diversity and population structure of Magnaporthe grisea, the causal agent of foxtail millet blast using microsatellites"

_PeerJ, doi:10.7717/peerj.16258_

## Round 0.1 · original submission · Major Revisions

Dear Dr. Tharmalingam,

Thank you for submitting your submission to PeerJ and we appreciate your patience for waiting.

Your manuscript had reviewed by three experts in your research areas. Based on their comments, and in my opinion for your article - Analysis of genetic diversity and population structure of Magnaporthe grisea, the causal agent of foxtail millet blast using microsatellites - that it requires a number of MAJOR REVISIONS. The main concerns were small sample sizes used for population genetic analysis. I would appreciate greatly it if you could response point-by-point and submit your revisions at your earliest possible time. Thank you.

Best regards,

Sincerely,

Tika Adhikari

·

Basic reporting

The authors showed the genetic diversity analysis of Magnoporthae griesa pathogen infecting foxtail millet in Tamil Nadu, India
This is necessary research that could improve foxtail millet breeding. However, I have a few concerns and comments for this MS as suggested below

1. The MS needs professional English editing to meet the publication criteria
2. The figures in the MS are not professional and need polishing to match scientific publication
3. IT would be great to indicate the regions of the isolates in Figure 3 and Figure 4
4. The discussion is very weak and needs improvement.
5. It would be better to reorganize and combine the supplementary figure 1 and 2 with Figure 2, to show the incidence and fungal spore phenotypes

Please correct the typo as indicated in the attached file
Line 45, close the parenthesis

Experimental design

No comment

Validity of the findings

No comments

Additional comments

It would be better to classify the pathogenic population based on the region they isolated. The difference and grouping based on plains and hills and how the genetic variation has occurred among these populations?

Reviewer 2 ·

Basic reporting

Line31-32 As a clustering analysis tool, PCoA should clarify its clustering results rather than describing the method itself.
Line147-150 Maximum likelihood method for constructing evolutionary trees appears to be more accurate and persuasive.
line209-220 The names of the groups, Group1 and Group2, need to be placed at their corresponding positions on the evolutionary tree, and a complete evolutionary tree should include scale bar.
line233-237 The author should display the name of the sample at the tip of the evolutionary tree, like in Fig 3. In this evolutionary tree, does cluster 3 correspond to group I in Fig 3? and the full paper does not explain why there are differences in the evolutionary trees established by ITS and SSR.Why these two isolates separated from the other isolates was not given a reasonable explanation by the author later.
line241 What is “adhoc”,maybe indicate the full name?and “p” in line254
line248-249 Do these three isolates correspond to Group1 in Fig3 or cluster3 in Fig4 ? The article lacks coherence and doesn't have a good connection, it just provides a general description of the results.
line250-252 The author should have explained gene flow in the discussion section, why it exists, whether it is due to the proximity of the regions? However, there was no such explanation given.
In Fig 5b, A complete graph should include titles for the axes.

Experimental design

no comment

Validity of the findings

no comment

Additional comments

The abstract should be restructured to highlight the novelty of the research rather than simply describing the methods and results.

Reviewer 3 ·

Basic reporting

The authors did a comprehensive work on the isolation and characterization of the MG isolates from FM. Introduction and Materials and methods sections were written with all relevant details, however, discussion section requires major revision.
The comments related to the improvement of discussion section is given in the annodated PDF file.

Experimental design

Standard methodology and analyses were carried out and presented.
The labelling of isolates in the figures would have been better for the readers to understand easily the grouping/diversity of isolates. The comments given in the annodated PDF file should be addressed.

Validity of the findings

The findings are novel to the location of the study. However, it should have been better discussed in the discussion section. Results are repeated in the discussion which requires some critical intrepretation of genetic diversity and its relation to disease incidence and location of collection etc.

Additional comments

Overall, the paper requires major revision before acceptance.

Annotated reviews are not available for download in order to protect the identity of reviewers who chose to remain anonymous.

---

## Round 0.2 · accepted · Accept

Dear Dr. Tharmalingam,

I am writing to inform you that your manuscript - Analysis of genetic diversity and population structure of Magnaporthe grisea, the causal agent of foxtail millet blast using microsatellites - has been accepted for publication.

Congratulations!


Sincerely,

Tika Adhikari

·

Basic reporting

The revised MS has improved and addressed all the concerns raised.

However, there are few minor errors, such as sentence alignment in Table 2 and check the sample location once again as given in the Table 1

Experimental design

Fine

Validity of the findings

The findings are validated well after revision